# Explaining placebo effects in an online survey study: Does 'Pavlov' ring a bell?

Rosanne M. Smits[1,2,3]*, Dieuwke S. Veldhuijzen[1,2,3], Tim Olde Hartman[4], Kaya J. Peerdeman[1,2], Liesbeth M. Van Vliet[1,2], Henriët Van Middendorp[1,2], Ralph C. A. Rippe[5], Nico M. Wulffraat[3], Andrea W. M. Evers[1,2,6]

1 Health, Medical and Neuropsychology unit, Leiden University, Leiden, The Netherlands, 2 Leiden Institute for Brain and Cognition, Leiden, The Netherlands, 3 Pediatric Immunology and Rheumatology, Wilhelmina Children's Hospital, Utrecht, The Netherlands, 4 Department of Primary and Community Care, Radboud Institute for Health Sciences, Radboud University Medical Center, Nijmegen, The Netherlands, 5 Research Methods and Statistics, Institute of Education and Child Studies, Leiden University, Leiden, The Netherlands, 6 Department of Psychiatry, Leiden University Medical Center, Leiden, The Netherlands

* R.M.Smits@fsw.leidenuniv.nl

## Abstract

### Objectives

Despite the increasing knowledge about placebo effects and their beneficial impact on treatment outcomes, strategies that explicitly employ these mechanisms remain scarce. To benefit from placebo effects, it is important to gain better understanding in how individuals want to be informed about placebo effects (for example about the underlying mechanisms that steer placebo effects). The main aim of this study was to investigate placebo information strategies in a general population sample by assessing current placebo knowledge, preferences for different placebo explanations (built around well-known mechanisms involved in placebo effects), and attitudes and acceptability towards the use of placebo effects in treatment.

### Design

Online survey.

### Setting

Leiden, The Netherlands.

### Participants

444 participants (377 completers), aged 16–78 years.

### Main outcome measures

Current placebo knowledge, placebo explanation preferences, and placebo attitudes and acceptability.

**Data Availability Statement:** The data underlying this study are available on DataVerseNL (https://doi.org/10.34894/XWKZSN).

**Funding:** This work was supported by grants of the Dutch Arthritis Foundation; the European Research Council (ERC Consolidator Grant ERC-2013-CoG-617700), and the Dutch Organization for Scientific Research (NWO-Vici grant 01 6.V I CL770. L52).

**Competing interests:** The authors declare no conflict of interest.

## Results

Participants scored high on current placebo knowledge (correct answers: $M$ = 81.15%, $SD$ = 12.75). Comparisons of 8 different placebo explanations revealed that participants preferred explanations based on brain mechanisms and positive expectations more than all other explanations ($F$(7, 368) = 3.618, $p$ = .001). Furthermore, attitudes and acceptability for placebos in treatment varied for the type of the condition (i.e. more acceptant for psychological complaints) and participants indicated that physicians do not always have to be honest while making use of placebo effects for therapeutic benefit.

## Conclusion

Our results brought forth new evidence in placebo information strategies, and indicated that explanations based on brain mechanisms and positive expectations were most preferred. These results can be insightful to construct placebo information strategies for both clinical context and research practices.

## Introduction

A substantial amount of literature has demonstrated the significant role of placebo effects and their positive influence on treatment outcomes [1–4]. Placebo effects refer to the beneficial effects after administration of an inert treatment or as an additive effect upon active treatments [5]. The underlying mechanisms involved in placebo effects have become increasingly well-understood and encompass learning mechanisms such as classical conditioning, instructional learning, and social observational learning [6–8]. Other factors that contribute to placebo effects involve the patient-physician relationship, communication styles, and trust [9]. Neuro-biological activation related to placebo effects (i.e. placebo-induced activation of specific brain regions) have demonstrated important insights into bodily responses after placebo administration [8]. Studies that integrate this wealth of study findings into clinical practice remain scarce, and information strategies that facilitate placebo effects in treatment are limited [1, 3, 5]. To optimize placebo information strategies, it is important to gain a better understanding in how individuals want to be informed about placebo effects (for example, about the underlying mechanisms that steer placebo effects), before this could be used for therapeutic benefit in clinical context.

Translating current knowledge of placebo effects into useful placebo information strategies can be challenging. According to previous studies that focused on attitudes and acceptability about placebo use in patients, several misconceptions exist [10–16]. For example, there seems to be a lack of understanding in what placebo and placebo effects entail. A telephone survey of 853 patients with chronic health problems found that 80.7% were familiar with the term 'placebo', whereas only about half of the participants (51.5%) had heard of the term 'placebo effects' [15]. A misconception regarding terminology also seems to exist. A focus group study of primary care patients indicated that the word 'placebo' was frequently associated with 'ineffective', which can be challenging when attempting to integrate placebo effects in treatment [11]. Moreover, attitudes towards acceptability of placebos in treatment are divided. In one study, 50–84% of the participants indicated that if they were informed about the potential benefit of placebo effects in treatment, they deemed placebo treatment acceptable [15]. However, data from other studies reported more nuances in attitudes, where participants reported this

to be strongly dependent on factors such as the type of a condition [11, 17]. Another recurrent topic from these patient studies was the need for transparency and shared decision-making [11, 14, 15]. Altogether, studies that investigate placebo attitudes and acceptability stress the need to develop placebo information strategies that are feasible in daily clinical practice and, more importantly, understandable for patients. Only then effective clinical implementation of placebo effects can take place.

Interestingly, the need for transparency is often contrasted by the (mis)conception that placebos solely work in a deceptive manner held by the majority of participants in several studies [11, 13, 15]. However, a growing body of literature demonstrates that even when participants are aware of placebo administration (i.e. open-label placebos), placebo effects are considerable [18–23]. In open-label designs, providing a comprehensive placebo rationale is essential, as this boosts (or induces) treatment effects when combined with placebos [24]. Several randomized controlled trials that implemented open-label placebos have shown clinically relevant outcomes in chronic low back pain, cancer-related fatigue, irritable bowel syndrome, major depressive disorder, attention deficit hyperactivity disorder, and allergic rhinitis [18–23]. However, it is remarkable that the majority of these studies have employed different explanations about placebo effects, ranging from classical conditioning [18, 19, 23, 25], the power of expectations [16, 18, 19, 23, 25, 26], neurobiological processes [19, 26], mind and body interaction [18, 20], or the use and efficacy of non-deceptive placebos [19, 26] (see S1 File for the explanations used in previous studies). Because of these variations, it is still unclear which of the explanations can be best used for daily practice, for example with open-label placebos.

This present study aims to investigate placebo information strategies in a general population sample to gain insights in future use for clinical practice. Because previous studies in this line of research have mainly focused on clinical populations [10–16], targeting a general population sample will be insightful as this population has not been influenced too much by specific clinical experiences and can therefore provide new insights in how placebo information strategies can be broadly implemented. First, this study examines the current knowledge of placebos and placebo effects. Secondly, different types of explanations about placebo effects based on their underlying mechanisms are assessed and are rated based on three outcomes; preference for each explanation, perceived efficacy for each explanation, and the willingness to participate in a treatment based on placebo effects for each explanation. Also, the study explores whether participants interpreted all different type of explanations as a single general underlying construct of placebo effects. In addition, for exploratory purposes this study investigated whether factors that have been associated with placebo effects in previous literature (i.e. age [27], gender [28, 29], education [16], dispositional optimism [30, 31], trait anxiety [32] neuroticism [33], beliefs about medication [34], and current placebo knowledge) could also have an impact on the preference for placebo information strategies Third and lastly, this study builds upon previous study findings by further exploring attitudes and acceptability towards the use of placebo effects in treatment [13, 15–17, 24, 35].

## Methods/Design

### Participants

Participants from the age of 16 years and older were recruited via social media (e.g., Facebook and WhatsApp) between April and June 2019. Participants had to be able to speak and understand Dutch. No further in- or exclusion criteria applied. In total, 444 participants started with the online survey, of which 377 participants completed the survey (see Table 1 for the demographic characteristics of the final sample). The first question of the survey contained the information letter and consent form. Participants that did not agree with the consent question,

**Table 1. Demographic characteristics (N = 377).**

| | |
|---|---|
| **Age** | 23 (20.5–28.0)[a] |
| Range (min and max) age in years | 16–78 |
| **Sex** | |
| Female | 240 (63.7)[b] |
| Male | 137 (36.3)[b] |
| **Educational level** | |
| Lower education[c] | 39 (10.3)[b] |
| Higher education[d] | 338 (89.7)[b] |
| **Medication used in the last month** | |
| Yes | 240 (63.9)[b] |
| No | 137 (36.1)[b] |
| **Medication use** | |
| Pain medication | 138 (36.6)[b] |
| Birth control | 38 (10.1)[b] |
| Medication for allergies/asthma/eczema | 19 (5)[b] |
| Other[e] | 45 (12.2)[b] |

[a]Median (IQR),

[b]N (%),

[c]primary and lower secondary education,

[d]higher general secondary education, pre-university education, higher vocational education, university and academic degree,

[e]Other: i.e., thyroid medication, insulin, antidepressants and ADHD/ADD medication).

were not able to continue with the survey and were not included in the study. Participants were compensated with a €6.50 monetary reward or two course credits for study participation. This study was approved by the Psychology Research Ethics Committee of Leiden University (CEP19-0204/53). Because this was the first study to compare placebo information strategies, there were no prior effect sizes for a power calculation. Instead, the sample size was based on previous placebo questionnaire studies. We therefore aimed to include 400 participants in line with comparable studies [13, 14, 17].

## Procedure

Participants were invited via a link in an e-mail or social media for an online survey via Qualtrics (https://leidenuniv.eu.qualtrics.com) entitled 'What do you know about placebo?'. The survey could be filled out on a mobile phone or computer and could be paused at any time. The first part of the survey focused on placebo effects and included three subsets presented in a fixed order: a PlaceboQuiz (to assess current placebo knowledge), placebo explanations (each explanation was rated based on their preference, perceived efficacy and willingness to participate in a treatment), and placebo scenarios (to assess acceptability for placebo use in different situations). This order was chosen so that current placebo knowledge was assessed first. After that, placebo explanations were presented in a randomized order to reduce the potential influence of carry-over effects coming from the information provided by the previous placebo explanations. The placebo scenarios were presented last because by then participants had received all placebo explanations and should be able to answer the questions independent of the current placebo knowledge they had. The second part of the survey focused on personal characteristics such as demographic factors, personality traits, and beliefs about medication

and medication use. At the end of the survey, participants were thanked for their participation and debriefed that the aim of this study was to gain a better understanding in how placebo effects are perceived and that this may help to optimize future treatment outcomes. The estimated time to fill out the entire online survey was about 45–60 minutes.

## Materials

**Placebo knowledge.**   A placebo quiz ('PlaceboQuiz') was included and consisted of 14 true or false statements about placebos and placebo effects (e.g., 'Placebos can trigger a physical response'). Items for this quiz varied in difficulty level from previous placebo surveys [10, 13–16, 26, 36] ranging in questions that were commonly answered correct (e.g., 'Thoughts can affect health') and questions that were commonly answered incorrect (e.g., 'Placebo effects do not work when a person knows he or she is taking a placebo'). Mean total correct scores from the PlaceboQuiz were calculated and percentage scores were computed ranging from 0–100%, where a higher score indicated more general knowledge about placebo effects.

**Placebo explanations.**   Different explanations about placebo effects were stated, which were all potential options for placebo information strategies. The explanations were comprised of a combination of previously used instructions in open-label studies and additional mechanisms involved in placebo effects [10, 13–17, 21, 23, 26, 36] (see S1 File for an overview of previously used explanations, and the explanations used in this current study). The explanations were based on 8 themes: classical conditioning, expectations, brain mechanisms, mind and body healing processes, social learning, trust, transparency, and finally a neutral explanation (which stated that placebo effects work for some people, not all, and that it is not entirely clear why). Eventually, all explanations provided three separate outcome scores: 1) *preference scores* on a numerical slider from 0–10 (indicating how much participants would like to receive each explanation), 2) *perceived efficacy*, indicating how effective participants perceived each explanation (5-point Likert scale, strongly disagree to strongly agree), and 3) *willingness to participate*, indicating the willingness to participate in a treatment based on each placebo explanation (5-point Likert scale, strongly disagree to strongly agree).

**Placebo attitudes and acceptability.**   Seven different situations of placebo use in a treatment context were provided that varied in the level of openness of a physician about placebo use (3 situations) and the extent of placebos integrated in treatment (4 situations). For example, one question about openness of placebo administration was "The physician only has to disclose placebo use afterwards and only when it works". For the integration of placebos in treatment, different methods of placebo use from previous literature were presented, for example as 'dose extenders' ("I think placebo treatment is acceptable when prescribed after a long period of medication") or 'therapeutic boosters' ("I think placebo treatment is acceptable when combined with another medical treatment") [20, 22, 37]. In addition to previous research about placebo attitudes indicating that attitudes may be dependent on factors such as the type of a condition [11, 15, 17, 38], answer categories were added for different types of complaints (i.e., in case of a) psychological complaints, b) a cold, c) chronic diseases, d) terminal diseases, e) never correct, or f) always correct). Participants could then choose for each statement for what type of condition they would deem placebo treatment appropriate. Multiple answers were possible.

**Demographic factors.**   Demographic information regarding age, sex, education level, and medication use were collected. Education was categorized according to the Dutch educational system in a lower (primary and lower secondary education) and a higher education level (general secondary education, pre-university education, higher vocational education, university and academic degree).

**Dispositional optimism.**   The Life Orientation Test-Revised (LOT-R) [39] assessed dispositional optimism and contained six self-report items (and four filler items) rated on a 5-point Likert scale ranging from 0 *(strongly disagree)* to 4 *(strongly agree)*, with total scores ranging from 0–24. Higher scores indicated more dispositional optimism [39].

**Neuroticism.**   The Revised NEO Personality Inventory (NEO-PI-R) [40] was used to assess neuroticism and consisted of eight items that were rated on a 5-point Likert scale ranging from 1 *(strongly disagree)* to 5 *(strongly agree)*. The scale resulted in total scores ranging from 8–40, with higher scores indicating a higher sensitivity to stressful situations [40].

**Trait anxiety.**   The Spielberger State Trait Anxiety Inventory (STAI) [41] was assessed to measure trait anxiety of participants. The STAI consisted of 20 items on a 5-point Likert scale ranging from 1 *(strongly disagree)* to 5 *(strongly agree)*. Total scores range from 20 to 80, with higher scores indicating higher trait anxiety levels [41].

**General attitudes towards medication.**   The General Attitudes towards Medication Survey (GAMQ) [42] was used to assess attitudes towards prescription medication in general, and consisted of 12 items on a 5-point Likert scale with scores ranging between 0 *(strongly disagree)* and 5 *(strongly agree)*. Final scores ranged from 0 to 60, with higher scores indicating a more positive attitude towards medication [42].

## Statistical analysis

Data was analyzed using IBM SPSS Statistics (version 25).

**Placebo knowledge.**   Mean percentage correct scores of the PlaceboQuiz were computed per item. Multiple regression analysis was conducted to analyze potential prediction of placebo knowledge by age, gender, education level, medication use, dispositional optimism, trait anxiety, neuroticism, and general attitudes towards medication.

**Placebo explanations.**   For the explanation comparisons, 8 placebo explanations were compared based on three outcome measures: preference scores, perceived efficacy scores, and willingness to participate scores. For preference scores, the outcomes were measured on a numerical scale and were entered in a repeated measures ANCOVA with the 8 explanations for each participant as the within-subject factor. The 8 explanations were entered in repeated measures ANCOVA as different timepoints to account for error variance within participants, since every participant had evaluated all 8 explanations. Furthermore, we added 8 placebo correlates: age, gender, education level, dispositional optimism, trait anxiety, neuroticism, placebo knowledge, and attitudes towards medication in the repeated measures ANCOVAs, treated as between-subject variables to explore whether these factors influenced preference scores for each explanation. Perceived efficacy scores and willingness to participate scores were measured on a Likert-scale and were first converted into numerical values using optimal scaling from non-linear principal component analyses (CATPCA) through the multiplication of individual scores with transformed values [43, 44]. After transformations, perceived efficacy scores and willingness to participate scores for the 8 explanations were entered into the same repeated measures ANCOVAs as described above.

For visual representation, the placebo correlates were categorized to compute simple slopes in low (-1 *SD*), mean, and high (+1 *SD*), or were categorized based on previous literature if available [39]. To measure whether the placebo information strategies were all interpreted by a single general underlying construct of placebo effects, linear principal component analysis (PCA) was conducted for preference scores and CATPCAs were conducted for perceived efficacy and willingness to participate scores with optimal scaling transformations in order to perform a principal component analysis that reproduces as much variance as possible based on the covariance matrix [43, 44].

**Placebo acceptability.** Percentages were computed to give an overview of which placebo scenarios are deemed acceptable.

We considered a significance level of $< .05$ to be significant for all analyses and conducted post-hoc Bonferroni corrections for multiple testing. Normality was tested based on skewness and kurtosis. Outliers were specified as exceeding a distance of 1.5 times the interquartile range from the adjacent quartile. For effect sizes, partial eta squared ($\eta p^2$) was reported with values of 0.01 considered as a small effect, 0.06 as a moderate effect and 0.14 as a large effect [45].

## Results

### Sample characteristics

Of 444 participants, 377 participants completed the survey. Data from 444 participants on the PlaceboQuiz, data from 401 participants on demographic factors and data from 377 participants of the entire survey were used for data analysis. For the analysis of placebo explanations, normality tests indicated an assumption violation. To test for the severity of this violation, we conducted sensitivity analyses to compare the directions of effects of the original dataset with square root transformations versus a dataset where data of outliers based on 1.5 interquartile range of the transformed data were excluded. Exploration of the 'outlier group' ($>1.5$ interquartile range) did not indicate differences from participant data within 1.5 interquartile range in demographic characteristics. Based on this finding, we included the data of the 'outlier' group in the analysis, as there were no clear indications why these responses were not plausible, or fitting to this sample, and chose to report the results of the complete dataset (N = 377, see S2 File for comparisons between the group within 1.5 interquartile range and the group that exceeded the 1.5 interquartile range).

### Placebo knowledge

Overall, the PlaceboQuiz indicated a mean of correct scores of 81.15% ($SD = 12.75$). The lowest score (22.5% correct) was found on the question stating that placebo effects could be induced without deception, and the highest scores (96.8%) were found on the questions stating that thoughts can affect health and that placebo effects do not only occur in alternative medicine. Scores on all questions are depicted in Table 2.

**Predictor analysis.** To assess potential predictors associated with placebo knowledge, a multiple regression analysis was conducted with age, gender, education level, medication use, dispositional optimism, trait anxiety, general attitudes towards medication, and neuroticism. Results showed that the regression model explained 13.6% of the variance, $F(8, 365) = 5.222$, $p < .001$, being mainly attributable to the significant relation between a higher education and more placebo knowledge (see Table 3).

### Placebo explanations

**Explanation comparisons.** The Huynh-Feldt correction was used to correct for sphericity violations for preference scores, perceived efficacy and willingness to participate. A significant difference in preference scores was found between explanations, $F(7, 368) = 3.618$, $p = .001$, $\eta p^2 = .010$. Bonferroni-corrected post-hoc tests for pairwise comparisons indicated that explanations based on brain mechanisms ($M = 6.91$, $SD = .09$, $p < .001$) and expectations ($M = 6.67$, $SD = .09$, $p < .001$) had significantly higher preference scores than all other explanations. The neutral explanation received significantly lower preference scores compared to the other explanations ($M = 4.28$, $SD = .12$, $p < .001$) (see Fig 1). No significant differences were found for the

**Table 2. Results PlaceboQuiz (N = 444).**

| | True | False | Correct answer |
|---|---|---|---|
| Placebo effects only occur in alternative medicine (such as acupuncture or herbal medicines) | 14 | 430 | False (96.8%) |
| Thoughts can affect your health | 430 | 14 | True (96.8%) |
| Positive expectations can have a positive effect on health | 429 | 15 | True (96.6%) |
| A pill that contains aspirin is called a 'placebo' | 19 | 425 | False (95.7%) |
| Trust in the physician and the prescribed treatment can add to placebo effects | 422 | 22 | True (95%) |
| Placebo effects are only present in psychological complaints (such as stress) | 27 | 417 | False (93.9%) |
| Placebo effects only occur in scientific research | 43 | 401 | False (90.3%) |
| A placebo can reduce symptoms such as pain | 372 | 72 | True (83.8%) |
| The packaging of a placebo (i.e. color of the pill) can influence its effects | 367 | 77 | True (82.7%) |
| Placebos are used to develop new drugs | 348 | 96 | True (78.4%) |
| Placebos can cause changes in the brain (such as producing chemical substances) | 345 | 99 | True (77.7%) |
| Placebos can induce a physical reaction | 325 | 119 | True (73.2%) |
| Placebos can also cause adverse effects | 233 | 211 | True (52.5%) |
| Placebo effects do not work when a person knows he or she is taking a placebo | 344 | 100 | False (22.5%) |

explanations based on perceived efficacy scores ($F(7, 367) = 1.645$, $p = .121$, $\eta p^2 = .004$ For willingness to participate, also no significant differences were found based on the explanations ($F(7, 368) = 1.119$, $p = .348$, $\eta p^2 = .003$).

**Analysis of placebo correlates.** *Preference scores*. Evaluation of the placebo correlates that were entered into the repeated measures ANCOVA as between-subject factors, indicated that preference scores differed based on age ($F(1,368) = 10.652$, $p = .001$, $\eta p^2 = .028$), education level ($F(1,368) = 12.363$, $p < .001$, $\eta p^2 = .033$), optimism ($F(1,368) = 11.461$, $p = .001$, $\eta p^2 = .030$), trait anxiety ($F(1,368) = 6.592$, $p = .011$ $\eta p^2 = .018$), and placebo knowledge ($F(1,368) = 6.606$, $p = .011$, $\eta p^2 = .018$), but not with gender, neuroticism, and attitudes towards medication. For example, participants who scored 1 SD higher on trait anxiety showed a stronger preference for the explanation based on positive expectations than participants lower on trait anxiety. Participants with higher optimism scores showed a stronger preference towards the explanation based on positive expectations than participants with lower optimism scores. Another example from the placebo correlates analysis shows that participants with lower education levels rated all explanations higher (with the exception of the explanation based on

**Table 3. Regression analysis summary for predictors of placebo knowledge.**

| Variable | B | SE | 95% CI | Beta | T | P |
|---|---|---|---|---|---|---|
| Age | -.07 | .05 | [-.17;.03] | -.07 | -1.34 | .182 |
| Gender | -.31 | 1.40 | [-.31; 2.43] | -.01 | -.22 | .826 |
| Education | 2.85 | .58 | [1.71; 4.00] | .25 | 4.9 | < .001 |
| Medication use | -1.09 | 1.36 | [-3.76; 1.58] | -.04 | -.80 | .422 |
| Optimism | -.31 | .17 | [-.65;.028] | -.09 | -1.80 | .072 |
| Trait anxiety | .03 | .15 | [-.26;.31] | .01 | .20 | .843 |
| GAMQ | -.43 | .72 | [-1.85;.99] | -.11 | -.59 | .553 |
| Neuroticism | -.09 | .14 | [-.37;.18] | -.04 | -.65 | .514 |

Note. GAMQ = general attitudes towards medication questionnaire.

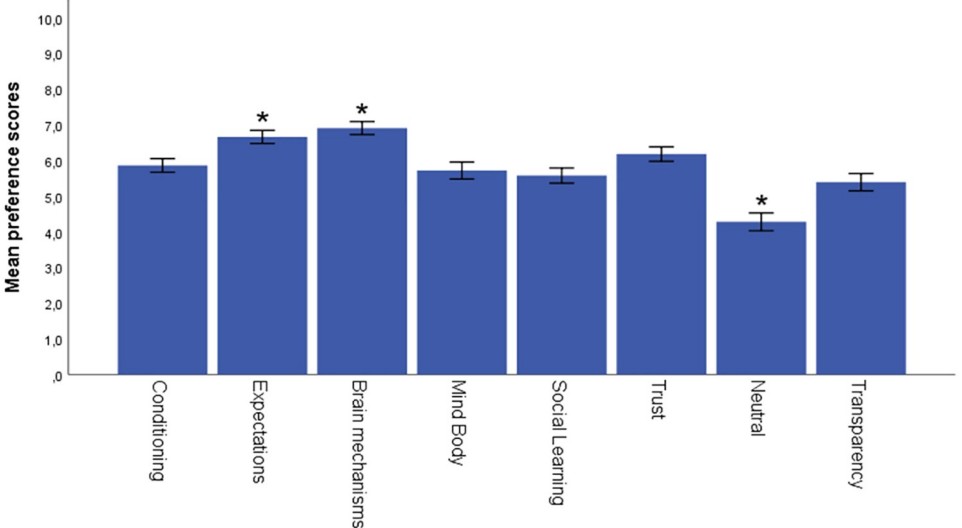

**Fig 1. Mean preference scores for all placebo explanations.** Error bars: 95% CI, *Preference scores for the explanations expectations, brain, and neutral significantly differed from the other explanations (p < .001).

brain mechanisms) than participants with higher education levels. See Fig 2 for a depiction of all significant variables of explanation preference scores.

*Perceived efficacy*. The placebo correlates that were entered into the repeated measures ANCOVA as between-subject factors.showed that perceived efficacy was associated with dispositional optimism ($F(1,367) = 9.172$, $p = .003$, $\eta p^2 = .024$), trait anxiety ($F(1,367) = 7.355$,

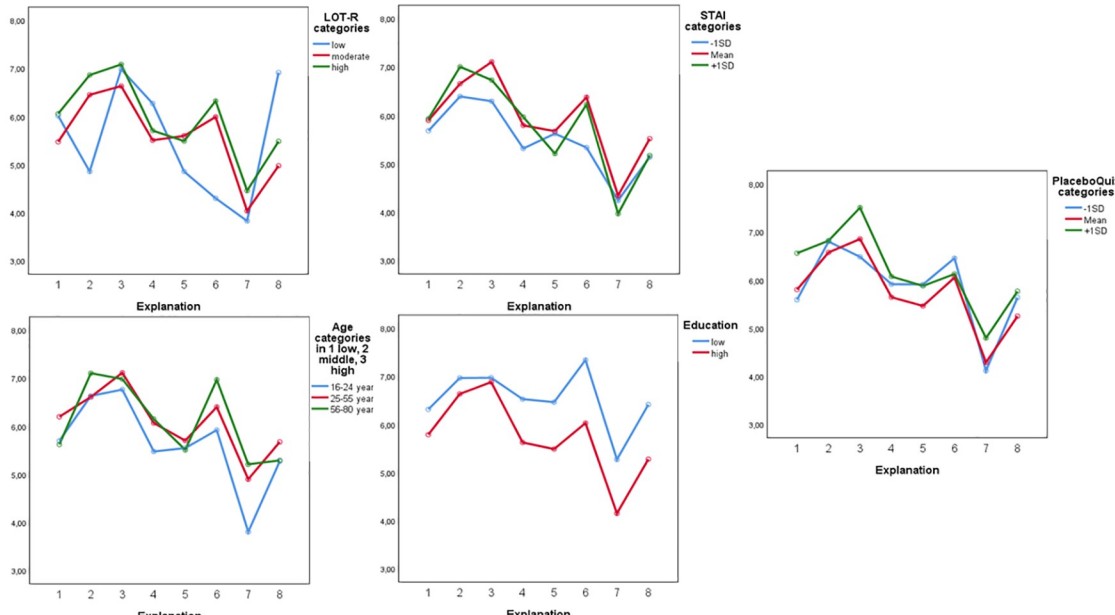

**Fig 2. Differential effects of significant placebo correlates: Optimism (LOT-R), trait anxiety (STAI), placebo knowledge (PlaceboQuiz), age, and education on preference scores for placebo explanations.** Explanations were numbered: 1 = conditioning, 2 = expectations, 3 = brain mechanisms, 4 = mind and body, 5 = social learning, 6 = trust, 7 = neutral, 8 = transparency. The y-axis represents mean preference scores.

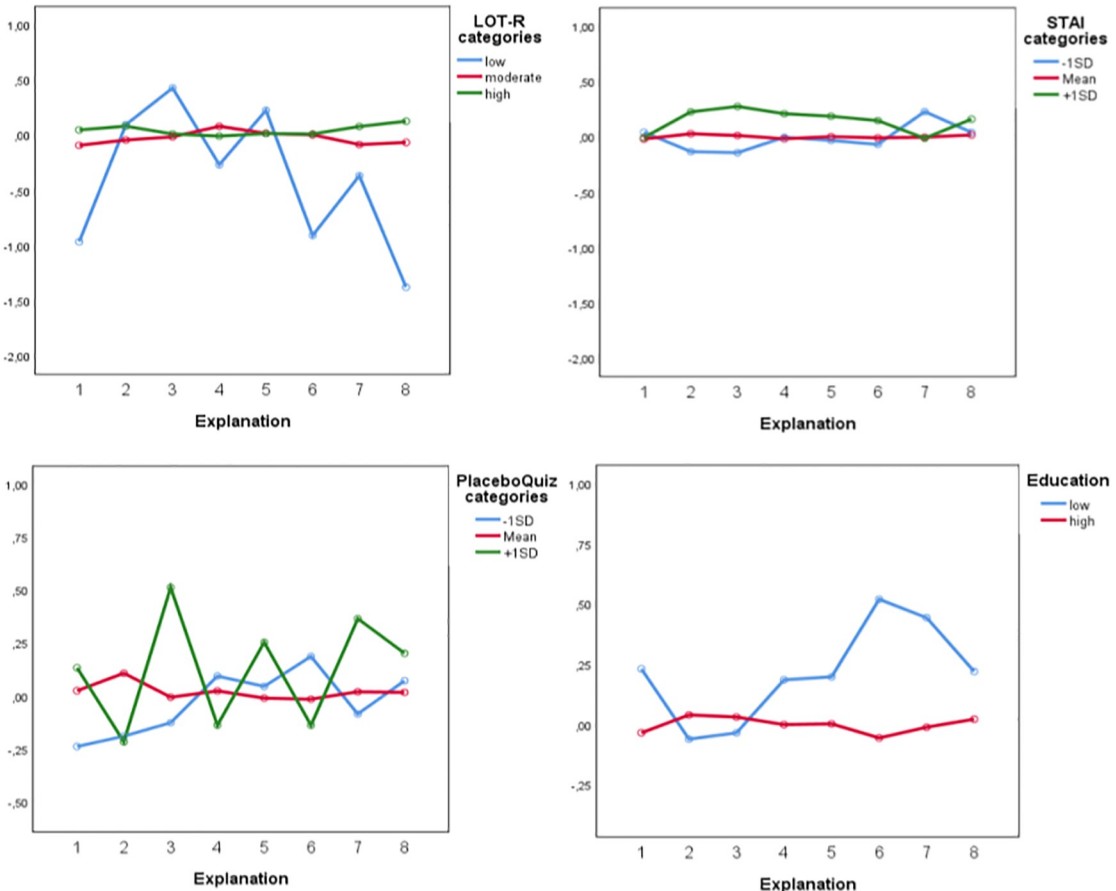

**Fig 3. Differential effects of placebo correlates for perceived efficacy: Optimism (LOT-R), trait anxiety (STAI), placebo knowledge, and education on perceived efficacy based on the placebo explanations.** Explanations were numbered (1 = conditioning, 2 = expectations, 3 = brain mechanisms, 4 = mind and body, 5 = social learning, 6 = trust, 7 = neutral, 8 = transparency). The y-axis represents transformed values of the perceived efficacy scores.

$p = .007$, $\eta p^2 = .020$), placebo knowledge ($F(1,367) = 5.136$, $p = .024$, $\eta p^2 = .014$), and level of education ($F(1,367) = 10.005$, $p < .002$, $\eta p^2 = .027$), indicating that the effectiveness of the explanations varied based on these factors. For example, participants who scored 1 SD higher on trait anxiety perceived almost all explanations as more effective than participants lower in anxiety. Participants with lower scores on dispositional optimism showed considerably more variation in how effective they perceived explanations, favoring the explanation based on expectations, than participants with moderate or high optimism scores who showed a more stable pattern across all explanations. In addition, participants with higher placebo knowledge also showed more variation than in the perceived effectiveness of the explanation, with the explanation based on brain mechanisms receiving the highest score for effectiveness (see Fig 3).

*Willingness to participate.* For willingness to participate placebo correlates trait anxiety ($F(1,368) = 12.835$, $p < .001$, $\eta p^2 = .034$) and general attitudes towards medication ($F(1,368) = 5.340$, $p = .021$, $\eta p^2 = .014$), indicated that the willingness to participate was dependent on the scores of these factors. For example, participants who scored 1 SD higher on trait anxiety showed more willingness to participate in placebo treatments based on explanations that focused on brain mechanisms and transparency, whereas lower scores on trait anxiety did not

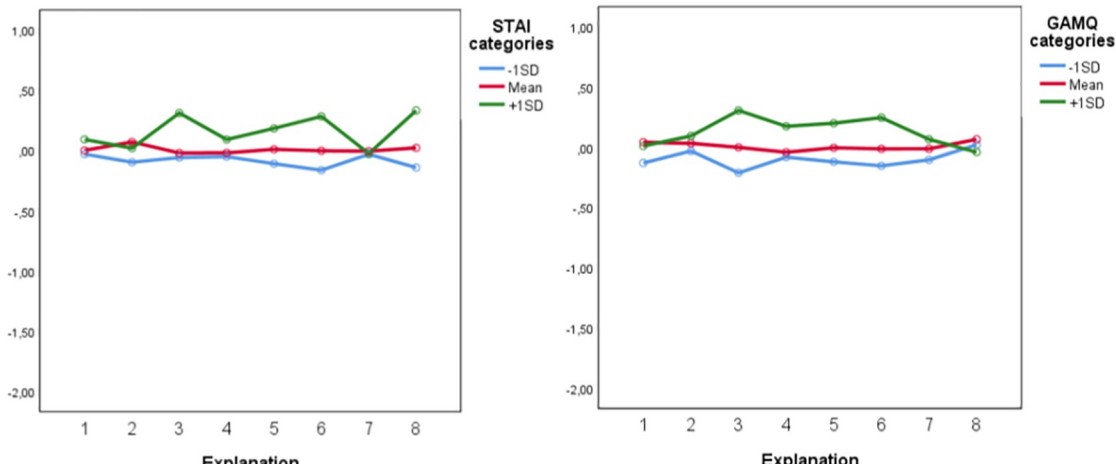

**Fig 4. Differential effects of placebo correlates: Trait anxiety (STAI), and general attitudes towards medication (GAMQ) on willingness to participate based on the placebo explanations.** Explanations were numbered (1 = conditioning, 2 = expectations, 3 = brain mechanisms, 4 = mind and body, 5 = social learning, 6 = trust, 7 = neutral, 8 = transparency). The y-axis represents transformed values of the willingness to participate scores.

show this difference. Furthermore, individuals with a more positive attitude towards medication showed more willingness to participate in placebo treatment based on the explanation of brain mechanisms (see Fig 4).

**Component analyses.** The PCA for preference scores revealed that all 8 explanations were interpreted as the same underlying construct, accounting for 36.24% of the total variance, and could not be reduced to a smaller set of interrelated explanations. For perceived efficacy, the CATPCA revealed a two-dimensional structure. Scores of two participants were excluded based on the scatterplot exploration because the scores did not fit either of the two components. Explanations based on expectations, mind and body, trust, social learning, and brain mechanisms described the first component and explained 19.68% of the variance. Explanations about conditioning, transparency and the neutral explanation were classified in a second component and explained 15.41% of the variance. Similar to the analysis for perceived efficacy, the CATPCA for willingness to participate revealed a two-dimensional structure for willingness to participate. Explanations based on expectations, mind and body, trust, conditioning and brain mechanisms described the first component and the explanations based on social learning, transparency and the neutral explanation accounted for the second component. The first component accounted for 22.74% of the variance and the second component for 15.81% of the variance. Because results from the component analyses indicated a dichotomy in interpretations for both outcomes of perceived efficacy and willingness to participate, additional analyses were conducted by repeating the abovementioned analyses to explore differences for the 2 underlying components of the 8 placebo explanations, however similar results were found (see S3 File).

## Placebo attitudes and acceptability

Participants were divided in their opinions about openness for the use of placebos in treatment, with equally high scores in the answer categories 'always correct' and 'never correct" for the different statements. For example, 19.2% of the participants indicated that it is always correct to use deception if the physician assumes that this will benefit the patient, whereas 22.8% of the participants indicated that this is never correct (see Fig 5). For the integration of placebos in treatment, highest scores were given when there is no other treatment available. The

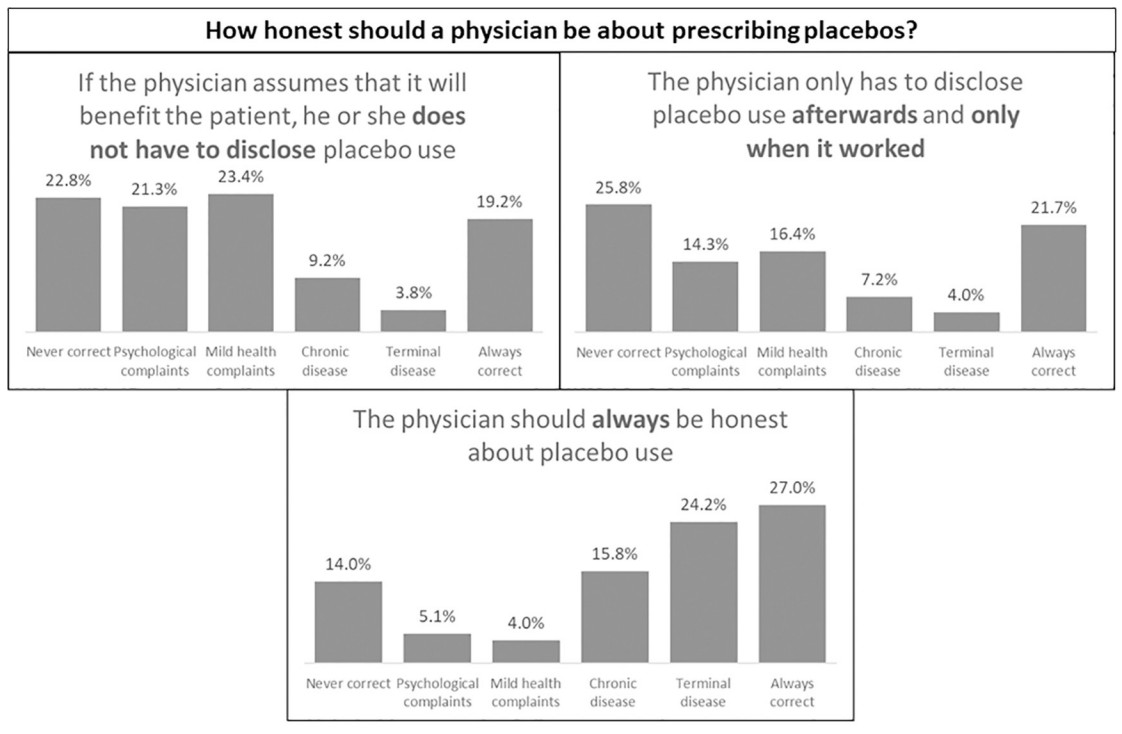

**Fig 5. Outcomes of placebo disclosure scores for all scenarios (N = 401).** Multiple answers were possible.

lowest scores were given when stated that placebos can never be combined with other treatments (see Fig 6). Furthermore, there was a clear distinction between conditions in acceptance for placebos in treatment or disclosure of placebos in treatment. Participants were most acceptant of placebo treatment when it comes to complaints of psychological nature and less acceptant towards placebo treatment for chronic or terminal diseases (see Figs 5 and 6).

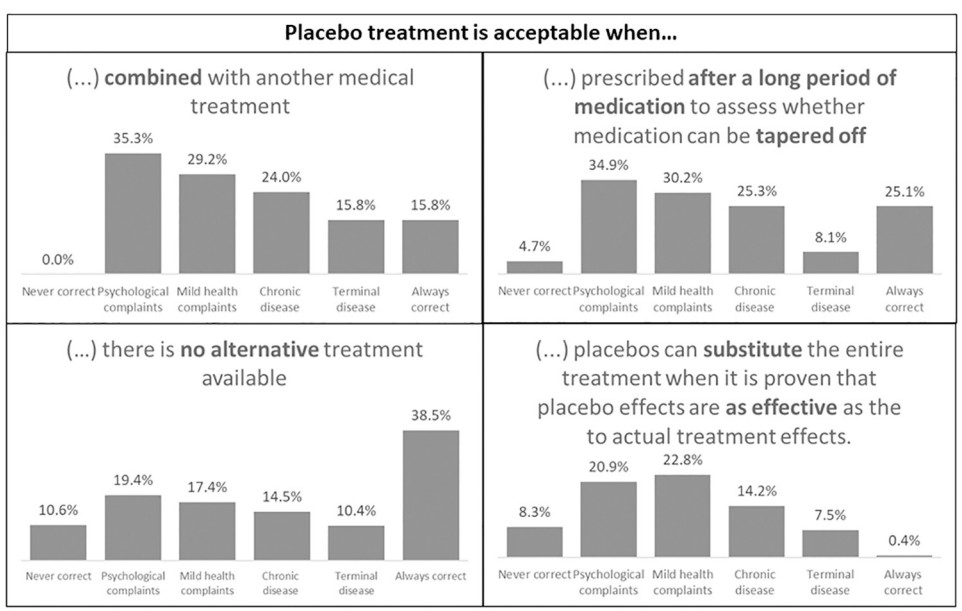

**Fig 6. Outcomes of placebo acceptability scores in different scenarios (N = 401).** Multiple answers were possible.

## Discussion

The primary aim of this study was to investigate placebo information strategies to gain insights in their potential use for clinical practice. First, this study examined the current knowledge about placebos and placebo effects. Results indicated that participants from our sample had an overall good understanding of both concepts, but that participants were less aware of the effectiveness of non-deceptive placebos and nocebo effects. Next, the different explanations about placebo effects based on their underlying mechanisms elucidated that explanations based on brain mechanisms and positive expectancies were most preferred, and that trait anxiety was a significant factor associated with all three explanation outcomes (preference scores, perceived efficacy, and willingness to participate). Moreover, attitudes and acceptability for placebos in treatment varied for the type of the condition (i.e. more acceptant for psychological complaints) and participants indicated that physicians do not always have to be honest while administering placebos.

In light of previous placebo literature, our findings show comparable results regarding overall placebo knowledge and showed that participants were generally well-acquainted with placebo effects, with the exception of non-deceptive placebos and nocebo effects [13, 14, 16, 17]. The results of the placebo correlates analysis tell us that factors previously associated to contribute to placebo effects, only explain a small portion of variance in placebo knowledge, and further research is warranted to explore this. Altogether, the results provide a clear direction for future placebo education to focus on non-deceptive placebo use and nocebo effects. Moreover, the present study adds to previous literature by demounting different placebo explanations from previous studies and assessing the preference for each explanation separately, to gain more insight in how placebo effects in treatments could be explained [16, 18, 19, 23, 25, 26]. Furthermore, compared with previous studies that focused on attitudes and acceptability about placebo use, this study was the first to our knowledge to examine the extent of placebos integrated in treatment and demonstrated comparable nuances for placebo acceptability for the type of a condition from previous literature [11, 17].

For a more in-depth understanding of whether participants interpreted all explanations as one underlying component, namely explaining the placebo effect, a component analysis was conducted. We considered this to be of importance in formulating placebo information strategies because a principal component analysis can serve as an exploration of the homogeneity of the reflection of internal beliefs about the underlying construct using the different formulations. For preference scores, results indicated that participant perceived the explanations as a single underlying construct, but when explanations were rated based on perceived efficacy and the willingness to participate in a treatment, a dichotomy was found. Explanations that were less positively stated (transparency, conditioning, social learning and the neutral explanation) were classified as one subcategory and more positively stated explanations (expectations, mind and body, trust, and brain mechanisms) were classified as a second subcategory. However, no significant differences in perceived efficacy or willingness to participate were found between the categories, indicating that some individuals preferred less positively stated explanations, for example individuals with relatively less positive attitudes towards medication (further explained in S3 File). Interestingly, the component analysis also indicated that all 8 placebo explanations only accounted for approximately one third of the variance, even though the explanations encompassed well-known mechanisms involved in placebo effects such as classical conditioning, expectations, neurological processes, social learning, trust, and open-label placebos [3, 8, 46, 47]. This finding raises the question whether participants may have preferred alternative explanations about placebo effects than the explanations used in the current study and previous literature, but further research is warranted to investigate this large amount of unexplained variance.

Besides investigating the internal interpretation of the placebo explanations with a principal component analysis (i.e. as a general underlying construct of placebo effects), this study also assessed whether personal factors contribute to the preference of a certain type of explanation. Our results demonstrated that trait anxiety was consistently associated with all three outcomes of placebo explanations (preference, perceived efficacy and willingness to participate). This indicates that participants that scored relatively highly on trait anxiety had a higher preference for explanations that involved positive expectations (i.e., the explanations about expectations and brain mechanisms), which could be a useful indicator for physicians that want to reassure anxious patients. However, in comparison to relevant norm groups (general population sample from Netherlands) we did find that mean trait anxiety scores from our sample were relatively higher ($M = 47.04$, $SD = 4.47$) than from the norm group ($M = 34.3$, $SD = 8.3$), so therefore results need to be interpreted with caution [48].

As the first study to our knowledge, the current sample consisted of participants from the general population. This group was chosen because this represents a large part of society that may be representative of a sample that occasionally visits the general practitioner and has not been influenced too much by specific clinical experiences. However, our sample had its limitations in regards to the distribution in age and education. Although the sample was rather large compared to previous placebo questionnaire samples (around 200 participants), most participants were highly educated and of relatively young age [13, 17]. Another limitation was the cross-sectional nature of the study, which prevents to assess whether attitudes towards the use of placebo effects can change over time. In order to evaluate whether our findings can be of potential use for clinical practice, a next step would be to target a patient sample or gain further insights from a sample with health care professionals. In addition, since this was one of the first studies that focused on specific placebo information strategies, this provides directions for future research. For instance, in order to utilize the current placebo information strategies it should also be investigated whether optimized placebo information strategies result in larger placebo effects. Lastly, it would furthermore be insightful to develop similar strategies for nocebo effects, since negative expectations can have detrimental effects on treatment outcomes [49]. Overall, results from our sample showed that participants were amenable towards placebo use in treatment, and provides further guidelines in regards to information strategies and knowledge gaps about placebo effects. These findings are insightful and may contribute to the development of placebo information strategies for future clinical implementation.

## Conclusion

This study provides new insights in how placebo effects can be explained, indicating that explanations based on brain mechanisms and positive expectations were most preferred in our sample to explain placebo effects. Moreover, our results brought forth insights when placebo use was deemed acceptable and encourages translation to clinical implementation. In addition, our results showed that even though our sample mainly consisted of higher educated participants, there was a lack of understanding about non-deceptive placebos and nocebo effects. These knowledge gaps are clear directions that need to be addressed to optimize placebo effects in treatments.

## Supporting information

**S1 File. Overview of placebo explanations based on their underlying mechanisms.**
(PDF)

**S2 File. Comparisons within 1.5 interquartile range and beyond 1.5 interquartile range.**
(PDF)

**S3 File. Additional analyses based on component scores for perceived efficacy and willingness to participate outcomes.**
(PDF)

**S4 File. Placebo questionnaire (Dutch version).**
(PDF)

**S5 File. Placebo questionnaire (English version).**
(PDF)

## Author Contributions

**Conceptualization:** Rosanne M. Smits, Dieuwke S. Veldhuijzen, Tim Olde Hartman, Kaya J. Peerdeman, Liesbeth M. Van Vliet, Henriët Van Middendorp, Ralph C. A. Rippe, Andrea W. M. Evers.

**Data curation:** Rosanne M. Smits.

**Formal analysis:** Rosanne M. Smits, Ralph C. A. Rippe.

**Funding acquisition:** Dieuwke S. Veldhuijzen, Andrea W. M. Evers.

**Investigation:** Rosanne M. Smits.

**Methodology:** Rosanne M. Smits, Dieuwke S. Veldhuijzen, Ralph C. A. Rippe, Andrea W. M. Evers.

**Project administration:** Rosanne M. Smits.

**Supervision:** Dieuwke S. Veldhuijzen, Andrea W. M. Evers.

**Visualization:** Rosanne M. Smits.

**Writing – original draft:** Rosanne M. Smits.

**Writing – review & editing:** Dieuwke S. Veldhuijzen, Tim Olde Hartman, Kaya J. Peerdeman, Liesbeth M. Van Vliet, Henriët Van Middendorp, Ralph C. A. Rippe, Nico M. Wulffraat, Andrea W. M. Evers.

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
