## [Decision Letter · Decision Letter 0]

10 Sep 2020

PONE-D-20-16802

Explaining placebo effects for clinical practice: Does ‘Pavlov’ ring a bell?

PLOS ONE

Dear Dr. Smits,

Thank you for submitting your manuscript to PLOS ONE. After careful consideration, we feel that it has merit but does not fully meet PLOS ONE’s publication criteria as it currently stands. Therefore, we invite you to submit a revised version of the manuscript that addresses the points raised during the review process.

Thank you for your submission. Based on my own reading of the manuscript, as well as the esteemed reviewers' suggestions, I would like to invite you to submit a revised version of this manuscript for reconsideration. In particular, I hope that this revision would allow you the opportunity to defend your manuscript, and address not only the methodological and statistical queries, but also the concerns that the manuscript may be overstating its importance and findings. While I believe that there is value in the contribution of this manuscript to the existing literature, I would strongly suggest that the conceptual grounding for this study, especially given the conceptual topic and framing re: placebo be better elucidated not only in the introduction but also discussion; furthermore I would suggest that the conclusions in relation to the results of this descriptive study be more tempered and realistic.

We look forward to receiving your revised manuscript.

Kind regards,

Haikel A. Lim, MD, MSc

Academic Editor

PLOS ONE

Journal Requirements:

3.  Please include additional information regarding the survey or questionnaire used in the study and ensure that you have provided sufficient details that others could replicate the analyses. For instance, if you developed a questionnaire as part of this study and it is not under a copyright more restrictive than CC-BY, please include a copy, in both the original language and English, as Supporting Information. If the survey instruments were translated, please indicate how the translation was carried out and validated.

5. Your ethics statement must appear in the Methods section of your manuscript. If your ethics statement is written in any section besides the Methods, please move it to the Methods section and delete it from any other section. Please also ensure that your ethics statement is included in your manuscript, as the ethics section of your online submission will not be published alongside your manuscript.

Reviewers' comments:

Reviewer's Responses to Questions

**Comments to the Author**

1. Is the manuscript technically sound, and do the data support the conclusions?

Reviewer #1: No

Reviewer #2: Yes

Reviewer #3: Partly

Reviewer #4: Yes

2. Has the statistical analysis been performed appropriately and rigorously? 

Reviewer #1: Yes

Reviewer #2: Yes

Reviewer #3: I Don't Know

Reviewer #4: Yes

3. Have the authors made all data underlying the findings in their manuscript fully available?

Reviewer #1: Yes

Reviewer #2: No

Reviewer #3: No

Reviewer #4: Yes

4. Is the manuscript presented in an intelligible fashion and written in standard English?

Reviewer #1: Yes

Reviewer #2: Yes

Reviewer #3: Yes

Reviewer #4: Yes

5. Review Comments to the Author

Reviewer #1: This is a study about the knowledge of and attitudes towards placebo effects in the general population. Technically it is well done but the problems are elsewhere. First, there are conceptual issues and second, the conclusions go to far and are not based on the results.

A) Conceptual issues:

1. Line 45. "Their potential use for clinical practice"?

2. Line 47. Use of placebos in clinical practice is completely different from harnessing placebo effects in care.

3. Line 60. "administering placebos". What does this actually refer to? Giving placebo pills, prescribing placebo pills (which is not possible in most countries), prescribing something that the physician does not believe to be pharmacologically active, prescribing something that the physician believes to be active but which is not, just being kind, etc. etc. are all very different situations.

4. Line 63. It is not clear what these results would mean for clinical practice.

5. Line 74-76. This is a very narrow understanding about placebo effects. They could also be seen as an essential component of any (positive) therapeutic relationship ("context and meaning effects").

6. Line 121-122. Research on open-label placebos is flourishing but it is not at all obvious that anything clinically useful would follow from that research. It will certainly increase our understanding of the phenomena but it is far too early to refer to their use in clinical practice.

7. Line 294 and page 16. I find three claims in the PlaceboQuitz conceptually problematic: "A placebo can reduce symptoms such as pain", "Placebos can induce a physical reaction", "Placebos can also cause side effects". This is common terminology, but, as the authors well know, placebos AS SUCH do not cause anything. This conceptual issue should at least be mentioned in the commenting text.

8. Line 432-433. Again, "optimize the potential use for clinical practice"???

9. Line 452-453. ""to examine the extent of placebos integrated in treatment" ???

10. Line 489. Again, "clinical implementation" meaning what?

B) The conclusions go too far.

11. Line 492. The normative conclusion that is drawn ("served best") does not follow from the descriptive results.

12. Line 493-494. The normative conclusion that is drawn ("encourages clinical implementation") does not follow from the descriptive results.

In sum, the study as such is technically done carefully. The authors should pay attention to the conceptual issues described above and write they conclusion in a far more modest way. This is a descriptive study and as such it is ok, but deriving conclusions about the justification of the use of placebos in clinical practice is a totally different issue.

Reviewer #2: The paper was able to achieve it's three main aims (1) it assessed the current knowledge of the general population about placebo/nocebo knowledge (2) assessed the different types of placebo explanations based on 3 outcomes and their association with different predictors and (3) assessed the acceptability of placebos in different clinical scenarios through a cross-section survey of the general population. It found areas where knowledge was lacking, found explanations that were favoured, and found opinions on when placebo use would be appropriate. Relevant recommendations were also given from the conclusions and would be useful knowledge to share with the scientific community.

This paper was an interesting read that investigated questions relating to how the placebo can be characterised to help provide information on how it can be translated into clinical therapies. Through its study design, this paper synthesised the different placebo explanations used and identified the heterogeneity in clinical placebo research currently. It also identified the different clinical scenarios it could be applied to, making the aims of the study clinically relevant and applicable.

Major comments:

(1) the study methods did not include a sample size calculation.

(2) there was little explanation or background given on why neuroticism, anxiety and optimism were used as predictors, or how they relate to the various placebo explanations. Hypotheses could have been generated regarding these relationships.

(3) the study population was the general population rather than a specific disease group, with the justification that this population has not been influenced by prior clinical experiences. However, the proposed use of placebos has been in diseased groups where traditional therapies have been ineffective, which would not be relevant to the general population that has not received prior clinical exposure. In addition, while medication use the study population was measured, comorbidities were not measured and controlled for.

Minor comments:

(1) in Table 2, row 1 of results, there is inconsistent reporting between the correct answer and the percentage answering with the correct answer.

(2) the axes in figures 2, 3, 4, 7 and 8 are difficult to identify.

(3) the conclusion that "attitudes and acceptability for placebos in treatment were mostly dependent on the type of the condition (i.e. more acceptant for psychological complaints)" in paragraph 1 of the discussion does not seem to be consistent with the results presented in figures 5 and 6.

Reviewer #3: Thank you for the opportunity to review this manuscript. This was a cross-sectional, survey-based study from Smits and colleagues at Leiden University in the Netherlands. The overall purpose was to characterize preference for various placebo information strategies for optimization in clinical practice. Specifically, the authors aimed to (1) examine current knowledge about placebo effects, (2) rate different types of explanations about placebo effects and their mechanisms based on preference, perceived efficacy, and willingness participate in a treatment based on a given placebo explanation, (3) extend prior work examining attitudes and acceptability towards placebos in treatment settings, (4) examine the underlying factor structure of their placebo explanations scale, and (5) examine the extent to which factors associated with placebo effects explained variance in explanation preferences.

Out of 444 total participants, 377 completed the survey. The primary findings are as follows:

(1) Placebo knowledge: Mean correct score of 81.1% on the placebo knowledge quiz with the lowest score observed for an item concerning whether deception is required in order to induce placebo effects (22.5 % correct)

(2) Placebo knowledge: 13.6% of variance in placebo knowledge was explained in the model. Education was the only significant predictor.

(3) Differences in placebo explanations

Preference: Participants differed in terms of preference for explanations with those based on brain mechanisms and expectations showing higher preference than others. The neutral explanation had significantly lower preference than the others.

Perceived efficacy: No differences

Willingness to participate: No differences

(4) Predictors of placebo explanations

Preference: Age, education, optimism, trait anxiety, and placebo knowledge

Perceived efficacy: optimism, trait anxiety, placebo knowledge and education

Willingness to participate: trait anxiety and general attitudes towards medication

(5) Factor structure

Preference: 1 factor explaining 36.24% of the variance

Perceived efficacy: 2 factors, 1 explaining 19.68% of the variance and 2 explaining 15.41% of the variance

Willingness to participate: 2 factors, 1 explaining 22.74% of the variance and 2 explaining 15.81% of the variance

(6) Placebo attitudes and acceptability:

Overall, I thought the study design was interesting and believe that this manuscript would add a unique perspective to the broader literature on placebo effects. However, the statistical analysis, results, and discussion sections in particular could benefit from more development. Thus, I would like to see your detailed response to my below comments before giving this manuscript any further consideration for publication.

Introduction-

123-139 – Consider making the order of your aims and results consistent for the reader. For instance, the aim to explore attitudes appears third in the list of aims whereas these data are presented last in the results section.

134-136 – Can you provide a stronger rationale for exploring the factor structure of your preference, perceived efficacy, and willingness to participate scales? How would this information benefit clinicians who want to use placebos in practice? As of now, I am having trouble understanding what this information adds to the overall manuscript and it seems more appropriate for a separate study of psychometric properties of their instrument (e.g., factorial validity)

Methods –

166-168 -You designed your study to control for order effects when collecting data on placebo explanations. Can you describe this a bit more? How many possible orders were available to a given participant? Did you confirm whether or not order had an effect on your results (e.g., does entering order as a covariate in your model affect the results)?

213-239- The investigators have used several standardized scales to help characterize the participant sample (i.e., Life Orientation Test-Revised, Revised NEO Personality Inventory, Spielberger State Trait Anxiety Inventory, General Attitudes towards Medication Survey). If available, please consider providing norms for each of these scales to help the reader contextualize whether the study sample is above or below average on each of these scales.

240- Can the authors comment on the statistical power of their study? Was an a priori power analysis conducted for any of the aims?

248-260- It is unclear to me why a repeated-measures model is being used here. Wouldn’t a one-way ANOVA/ANCOVA be sufficient since the overall design of the study did not involve multiple timepoints?

Results -

301-307 – Please provide the results for the overall regression model. Was the overall model significant? This is critical for interpreting the rest of your findings. In other words, if the overall model was not significant, observing individual predictors that were significant (e.g., education) does not really matter.

328-381- The authors describe results of three different regression models presumably examining predictors of preference, perceived efficacy, and willingness to participate, but these models are not described in the statistical analysis section. What kind of regression model was used? How was the dependent variable score treated? For each model, was the score a sum total from the various items used to characterize preference, perceived efficacy, and willingness to participate? This will definitely need more detail.

329-338- Please provide the results for the overall regression model in addition to your summary of individual predictors. Was the overall model significant?

351-362- Please provide the results for the overall regression model in addition to your summary of individual predictors. Was the overall model significant?

372-381- Please provide the results for the overall regression model in addition to your summary of individual predictors. Was the overall model significant?

Discussion-

444-446- The investigator appears to interpret “Placebos can also cause side effects” as knowledge about nocebo effects. Indeed, information about nocebo effect knowledge is very important, but I am not entirely convinced that is what they measured with this item and would urge the authors to exercise some caution with referring to it as such. How do we know the respondents are assigning a negative connotation to the term “side effects”? It is unfortunate that this was not spelled out more clearly for study participants (e.g., Placebos can also cause negative or harmful effects).

444-447 – There’s hardly any discussion of the regression model that explored predictors of placebo quiz performance and found 13.6% variance explained. Was the amount of variance explained lower or higher than the authors expected? What other factors could account for the large amount of variance that was not explained by the model?

456-474 – I found the exploration and discussion of the underlying factor structure to be rather distracting and I am not sure this adds much value to the paper. In your revisions, please either make this more apparent to the reader or consider removing altogether.

479-480 – “Although the sample was rather large (377 completers)…”. Relative to which specific studies?

486-489 – Despite listing this as one of their aims, there’s also very little discussion of how exactly their results have extended prior work examining attitudes and acceptability towards placebos in treatment settings. I think this could be a separate paragraph rather than squeezed in as an afterthought in it’s present place in the manuscript.

Limitations/Future directions section –

The discussion of the study limitations is pretty thin. Please add a paragraph or two on the potential shortcomings of this study to help the reader contextualize the findings. A few things that come to mind are:

(1) The cross-sectional nature of the study (e.g., what if attitudes change over time as new experiences occur (e.g., clinical trial participation, patient-provider interactions?). This is especially relevant considering how young your sample is.

(2) How might various populations with chronic mental or physical health conditions benefit from this information? Is the present data sufficient for these groups or is more data directly collected from patient populations needed?

(3) Is there anything to be learned from applying this research question/approach to nocebo effects and more thoroughly characterizing knowledge of nocebo effects than what was done here?

(4) Just because a participant prefers a given strategy does not necessarily mean this translates to the largest placebo effects. This of course is an empirical question that could be tested in an experimental design and merits some thought/discussion about how such a study could be conducted.

To be clear, these are just some potential ideas worth considering and I’m not saying that each of these needs to be added as limitations. Rather, I would like to see you give some thoughtful consideration to what you think are the primary limitations and discuss them accordingly.

Reviewer #4: The authors conducted an online survey to examine knowledge and attitudes of the general population towards placebos, placebo effects, and placebo mechanisms. Results indicate that knowledge about placebo effects is generally high, with better knowledge in well-educated participants. Furthermore, the participants preferred explanations of placebo effects based on brain mechanisms and expectations, and preferences varied with age, education level, optimism, trait anxiety, and placebo knowledge. Perceived efficacy of the 8 proposed explanations showed comparable associations, while the willingness to participate varied with trait anxiety and attitudes towards medication. PCA revealed a two-dimensional structure for perceived efficacy and willingness to participate, separating between more positively stated mechanisms (such as expectations/mind-body/trust/brain mechanisms), and more negatively stated ones (such as transparency/neutral explanation). The use of placebos in clinical pratice was widely accepted.

The study is well designed and reported, the statistical analysis is sound and the results support the conclusions. The findings are important as they provide various approaches for implementing placebo effects in clinical practice. The following issues should be addressed before publication:

- The sample comprises mainly young well-educated people. This is a major limitation of the results, since older and less-educated people may have different knowledge and attitudes about placebos and placebo effects. The authors discuss this issue very shortly. The implications are important and far-reaching and need to be addressed much more carefully by the authors. Furthermore, details of recruitment need to be explained in more detail (e.g., to whom the emails were sent, which social media platforms were used, etc.), and a rationale should be provided why the authors did not chose a more representative sample.

- The title is very vague. It should include the type of study as well as the restriction to younger adults.

- In Table 2, the entries for the first items should be double-checked; numbers appear to be reversed.

6. PLOS authors have the option to publish the peer review history of their article (what does this mean?). If published, this will include your full peer review and any attached files.

Reviewer #1: **Yes: **Pekka Louhiala

Reviewer #2: No

Reviewer #3: **Yes: **Jacob B. Lindheimer

Reviewer #4: No

---

## [Author Response · Author response to Decision Letter 0]

3 Dec 2020

Responses to the reviewers:

Reviewer #1:

This is a study about the knowledge of and attitudes towards placebo effects in the general population. Technically it is well done but the problems are elsewhere. First, there are conceptual issues and second, the conclusions go too far and are not based on the results.

Our response: We thank the reviewer for this important comment. We agree with the reviewer and have made revisions throughout the manuscript. Please see below for more detailed explanations. 

A) Conceptual issues:

1. Line 45. "Their potential use for clinical practice"?

 Our response: We have now adjusted this to “gain insights” instead of “optimize”, with ‘their’ referring to placebo information strategies that have been investigated in the current study. With ‘for clinical practice’ we aimed to create insight in placebo strategies that could further be developed for health care professionals (l. 44-46). Based on the valid remark of the reviewer we have also chosen to drop ‘for clinical practice’ from the title. 

2. Line 47. Use of placebos in clinical practice is completely different from harnessing placebo effects in care. 

 Our response: In line 48 we have addressed this by specifically stating that we are interested in placebo effects (not specifically and solely the administration of placebos itself), thereby underlining the relevance of investigating placebo information strategies by our three core objectives: 1) assessing current placebo knowledge, 2) preferences for different placebo explanations (built around well-known mechanisms involved in placebo effects), 3) and attitudes and acceptability towards the use of placebos and placebo effects.

3. Line 60. "administering placebos". What does this actually refer to? Giving placebo pills, prescribing placebo pills (which is not possible in most countries), prescribing something that the physician does not believe to be pharmacologically active, prescribing something that the physician believes to be active but which is not, just being kind, etc. etc. are all very different situations. 

 Our response: Thank you for this insightful remark; we have now changed this to making use of placebo effects for therapeutic benefit (l. 62). However, it is important to note that the questions from the attitudes and acceptability scales also focused on administering placebos itself because we were particularly interested about attitudes towards the deceptive nature of placebo use. This was more of a hypothetical question indeed, because in many countries such as The Netherlands where this study was conducted the use of deception in treatment is prohibited.

4. Line 63. It is not clear what these results would mean for clinical practice. 

 Our response: We understand your concern and have now addressed this by emphasizing that this study may gain insights for future development of placebo information strategies, instead of focusing on the direct use of these for clinical practice (l. 65-66). The reason why our results may be particularly insightful for clinical practice is because health care professionals may use the descriptions and examples from this study to explain placebo effects, and form this study also gained insights into what explanations are preferred more than others. Furthermore these insights may facilitate further development of placebo information strategies. We addressed this in a more nuanced manner in the last paragraph of the abstract and discussion (l. 531-541). 

5. Line 74-76. This is a very narrow understanding about placebo effects. They could also be seen as an essential component of any (positive) therapeutic relationship ("context and meaning effects").

 Our response: We agree with this notion that placebo effects encompass a broad range of effects including contextual and meaning effects, however for this manuscript we have chosen to use the general explanation for placebo effects which has been established by expert consensus (Evers et al., 2018). The reason for this was also because this manuscript focused on the treatment benefits that could be induced by developing placebo information strategies, and this definition highlights this as well (l. 78).

6. Line 121-122. Research on open-label placebos is flourishing but it is not at all obvious that anything clinically useful would follow from that research. It will certainly increase our understanding of the phenomena but it is far too early to refer to their use in clinical practice. 

 Our response: We agree with the concerns raised by the reviewer on translating OLP results conducted with healthy controls to clinical settings. However, in our overview we have mainly focused on open-label placebo effects that demonstrated meaningful effects in clinical samples. Conversely, placebo information strategies are of particular relevance for OLP studies, since these harness the desired effect of OLP and provide a rationale to patients/participants why these could induce beneficial effects (l. 116-124). 

7. Line 294 and page 16. I find three claims in the PlaceboQuiz conceptually problematic: "A placebo can reduce symptoms such as pain", "Placebos can induce a physical reaction", "Placebos can also cause side effects". This is common terminology, but, as the authors well know, placebos AS SUCH do not cause anything. This conceptual issue should at least be mentioned in the commenting text. 

 Our response: As we understand your concern, we have also aimed to make the PlaceboQuiz, and the rest of the questionnaire, as readable as possible for participants. It would perhaps have been more scientifically correct to state this as “A placebo, when combined with positive expectations, can reduce symptoms such as pain”, for example. However, we had to make a decision based on the trade-off of being scientifically accurate and user friendliness/readability. Additionally, we did not want to explain too much about placebo effects since this part of the questionnaire was aimed to investigate current knowledge about placebo effects, and explaining placebo effects would interfere with this. Lastly, the majority of the questions were retrieved from previous placebo surveys from different samples (l. 193-194).

8. Line 432-433. Again, "optimize the potential use for clinical practice"??

 Our response: We have adjusted this in line with the first comment; “gain insights” instead of “optimize” (l. 464-465).

9. Line 452-453. ""to examine the extent of placebos integrated in treatment" ??? 

 Our response: This followed from the attitudes and acceptability scales, where questions were formulated more hypothetical to explore attitudes towards acceptance. In previous placebo studies, placebos were integrated as dose extenders (e.g. Sandler, Glesne & Bodfish, 2010) or to boost therapeutic effects (e.g. Ader et al., 2010). To gain insight into the acceptability about how placebos could be integrated in actual treatment regimens, we therefore asked participants their attitudes on this. The answer categories are also depicted in Fig 6: ‘Placebo treatment is acceptable when.. 1) combined with another medical treatment (‘booster’), 2) prescribed after a long period of medication (‘dose extender’), 3) when there is no alternative treatment, 4) placebos can substitute the entire treatment when proven equally effective.’ To make this more clear to the reader, we addressed this rationale in the method section (under Placebo attitudes and acceptability): “For the integration of placebos in treatment, different ways of using placebos were presented, for example as ‘dose extenders’ (“I think placebo treatment is acceptable when prescribed after a long period of medication”) or ‘therapeutic boosters’ (“I think placebo treatment is acceptable when combined with another medical treatment”), in line with previous methods of placebo use” (l. 220-227). 

10. Line 489. Again, "clinical implementation" meaning what? 

 Our response: Clinical implementation in the sense that this study exposed knowledge gaps that may also be present in clinical samples, and therefore may be useful for health care providers to keep these gaps in mind when using or explaining placebo effects. Moreover, it addressed attitudes towards acceptability of the use of placebo effects; these findings are necessary to understand how placebo effects can be used (in research but also for clinical practice). As we understand and agree with your concern to translate our findings from a general population sample to clinical interpretations, we have addressed this by adding more nuances in the text such as ‘gaining insights for clinical practice’ instead of former statements such as ‘optimizing use for clinical practice’ and removed ‘for clinical practice’ from the study’s title (l. 532).

B) The conclusions go too far.

Our response: We agree with the reviewer and revised the conclusions described in the manuscript. Please see below for more detailed explanations. 

11. Line 492. The normative conclusion that is drawn ("served best") does not follow from the descriptive results. 

 Our response: We agree that this may have been an overstatement of our findings, since actual placebo effects were not measured in this study. We therefore corrected this to “most preferred in our sample”, which can be directly derived from our results (l. 551-552).

12. Line 493-494. The normative conclusion that is drawn ("encourages clinical implementation") does not follow from the descriptive results.

 Our response: We revised this to: “Our results brought forth insights when placebo use was deemed acceptable, which encourages translation to clinical implementation ” (l.552-554)

In sum, the study as such is technically done carefully. The authors should pay attention to the conceptual issues described above and write they conclusion in a far more modest way. This is a descriptive study and as such it is ok, but deriving conclusions about the justification of the use of placebos in clinical practice is a totally different issue.

 Our response: We thank the reviewer for the time and effort of reviewing our manuscript and for the compliments. We have now addressed the concerns raised by the reviewer in our revised manuscript We hope to have satisfactorily addressed the comments with this reply.

Reviewer #2: The paper was able to achieve it's three main aims (1) it assessed the current knowledge of the general population about placebo/nocebo knowledge (2) assessed the different types of placebo explanations based on 3 outcomes and their association with different predictors and (3) assessed the acceptability of placebos in different clinical scenarios through a cross-section survey of the general population. It found areas where knowledge was lacking, found explanations that were favoured, and found opinions on when placebo use would be appropriate. Relevant recommendations were also given from the conclusions and would be useful knowledge to share with the scientific community.

This paper was an interesting read that investigated questions relating to how the placebo can be characterised to help provide information on how it can be translated into clinical therapies. Through its study design, this paper synthesised the different placebo explanations used and identified the heterogeneity in clinical placebo research currently. It also identified the different clinical scenarios it could be applied to, making the aims of the study clinically relevant and applicable.

Our response: We thank the reviewer for the compliments on our study. 

Major comments:

(1) the study methods did not include a sample size calculation.

 Our response: Thank you for this comment. We have now included more information on our sample size calculation. This number (N=400) was based on samples from previous placebo questionnaires. We have added this information in lines 161-164.

(2) there was little explanation or background given on why neuroticism, anxiety and optimism were used as predictors, or how they relate to the various placebo explanations. Hypotheses could have been generated regarding these relationships.

 Our response: Because this was one of the first studies that explored the relation to placebo explanations, no specific hypotheses were formed. Instead, this study aim was of an exploratory nature which is explained further in lines 137-141.

(3) the study population was the general population rather than a specific disease group, with the justification that this population has not been influenced by prior clinical experiences. However, the proposed use of placebos has been in diseased groups where traditional therapies have been ineffective, which would not be relevant to the general population that has not received prior clinical exposure. In addition, while medication use the study population was measured, comorbidities were not measured and controlled for.

 Our response: Thank you for this comment. The study’s aim was to target the general population to gain a broad understanding of placebo knowledge, explanation preference, and attitudes towards acceptability. An interesting subsequent step would be to further investigate this in clinical populations. Based on your valid comment, we have chosen to alter the manuscript title and drop ‘for clinical practice’. We do not however agree with the comment of the reviewer that the proposed use of placebos is only in diseased groups where no traditional treatment options are left; it is well established that placebo effects are an inherent element of all traditional treatments and therefore more knowledge of the general population is certainly important.

Minor comments:

(1) in Table 2, row 1 of results, there is inconsistent reporting between the correct answer and the percentage answering with the correct answer.

 Our response: Thank you for pointing this out. This has now been corrected (l. 327).

(2) the axes in figures 2, 3, 4, 7 and 8 are difficult to identify.

 Our response: Numbers have been addressed in the figure captions.

(3) the conclusion that "attitudes and acceptability for placebos in treatment were mostly dependent on the type of the condition (i.e. more acceptant for psychological complaints)" in paragraph 1 of the discussion does not seem to be consistent with the results presented in figures 5 and 6.

 Our response: We agree with your concern and have now changed this to “varied for the type of the condition” (l. 471).

Reviewer #3: Thank you for the opportunity to review this manuscript. This was a cross-sectional, survey-based study from Smits and colleagues at Leiden University in the Netherlands. The overall purpose was to characterize preference for various placebo information strategies for optimization in clinical practice. Specifically, the authors aimed to (1) examine current knowledge about placebo effects, (2) rate different types of explanations about placebo effects and their mechanisms based on preference, perceived efficacy, and willingness participate in a treatment based on a given placebo explanation, (3) extend prior work examining attitudes and acceptability towards placebos in treatment settings, (4) examine the underlying factor structure of their placebo explanations scale, and (5) examine the extent to which factors associated with placebo effects explained variance in explanation preferences.

Out of 444 total participants, 377 completed the survey. The primary findings are as follows:

(1) Placebo knowledge: Mean correct score of 81.1% on the placebo knowledge quiz with the lowest score observed for an item concerning whether deception is required in order to induce placebo effects (22.5 % correct)

(2) Placebo knowledge: 13.6% of variance in placebo knowledge was explained in the model. Education was the only significant predictor.

(3) Differences in placebo explanations

Preference: Participants differed in terms of preference for explanations with those based on brain mechanisms and expectations showing higher preference than others. The neutral explanation had significantly lower preference than the others.

Perceived efficacy: No differences

Willingness to participate: No differences

(4) Predictors of placebo explanations

Preference: Age, education, optimism, trait anxiety, and placebo knowledge

Perceived efficacy: optimism, trait anxiety, placebo knowledge and education

Willingness to participate: trait anxiety and general attitudes towards medication

(5) Factor structure

Preference: 1 factor explaining 36.24% of the variance

Perceived efficacy: 2 factors, 1 explaining 19.68% of the variance and 2 explaining 15.41% of the variance

Willingness to participate: 2 factors, 1 explaining 22.74% of the variance and 2 explaining 15.81% of the variance

(6) Placebo attitudes and acceptability:

Overall, I thought the study design was interesting and believe that this manuscript would add a unique perspective to the broader literature on placebo effects. However, the statistical analysis, results, and discussion sections in particular could benefit from more development. Thus, I would like to see your detailed response to my below comments before giving this manuscript any further consideration for publication.

Our response: We thank the reviewer for the compliments on our study. We addressed the questions and concerns below. 

Introduction-

123-139 – Consider making the order of your aims and results consistent for the reader. For instance, the aim to explore attitudes appears third in the list of aims whereas these data are presented last in the results section.

 Our response: Thank you for this insightful comment, we have now changed the order of objectives in the introduction to be in line with the order of results (l. 135-143). 

134-136 – Can you provide a stronger rationale for exploring the factor structure of your preference, perceived efficacy, and willingness to participate scales? How would this information benefit clinicians who want to use placebos in practice? As of now, I am having trouble understanding what this information adds to the overall manuscript and it seems more appropriate for a separate study of psychometric properties of their instrument (e.g., factorial validity)

 Our response: Because this study was one of the first to specifically aim at formulating different placebo information strategies, the rationale for integrating a principal component analysis was to investigate whether participants perceived these explanations as the same underlying construct, i.e. in a homogeneous way. We considered this to be of importance in formulating placebo information strategies because a principal component analysis can serve as an exploration of the homogeneity of the reflection of internal beliefs about the underlying construct using the different formulations. This can be seen as additional information to for example Cronbach’s Alpha, which is a measure to reflect the overall consistency of these explanations as a reflection of internal beliefs, while assuming (!) firstly that these formulations are actually unidimensional and secondly that all scales are perfectly numerical.. To avoid these implicit assumptions, the current analysis is the only suitable alternative to address these issues within the estimations. To clarify this, we have adjusted this part in the method section to: “To measure whether the placebo information strategies were all interpreted by explaining a single general underlying construct of placebo effects”(l. 288-294).

Methods –

166-168 -You designed your study to control for order effects when collecting data on placebo explanations. Can you describe this a bit more? How many possible orders were available to a given participant? Did you confirm whether or not order had an effect on your results (e.g., does entering order as a covariate in your model affect the results)?

 Our response: Thank you for the possibility to explain this further., We controlled for potential order effects of placebo explanations in the conceptualization of the study design by presenting the explanations in a random order by the online survey tool that was used (Qualtrics). The rationale behind this was that previous explanations could have an effect on the explanations that followed. Unfortunately we have not collected data to control for potential order effects but the random presentation of the order will have minimized this potential effect.

213-239- The investigators have used several standardized scales to help characterize the participant sample (i.e., Life Orientation Test-Revised, Revised NEO Personality Inventory, Spielberger State Trait Anxiety Inventory, General Attitudes towards Medication Survey). If available, please consider providing norms for each of these scales to help the reader contextualize whether the study sample is above or below average on each of these scales.

 Our response: We have added this to our discussion (l. 518-521) with normative data for trait anxiety, as this was a consisted factor found in the analysis associated with all three outcomes (preference scores, perceived efficacy and willingness to participate).

240- Can the authors comment on the statistical power of their study? Was an a priori power analysis conducted for any of the aims?

 Our response: A sample size calculation was based on the average sample sizes from previous placebo questionnaire studies (N=400). We have now included this information in the method section (l. 161-164). Because this is the first study that compares placebo information strategies, no specific power calculation could be performed to target a specific effect. This sample size was indeed determined a priori as described in our ethical proposal for this study. 

248-260- It is unclear to me why a repeated-measures model is being used here. Wouldn’t a one-way ANOVA/ANCOVA be sufficient since the overall design of the study did not involve multiple timepoints?

 Our response: The rationale for a repeated measures ANCOVA instead of a between subject ANOVA was because all participants received the same eight explanations, and therefore within-subject errors could be reduced by comparing all eight explanations from participants to each other. A one-way ANOVA would consider the 8 explanations to be independent levels of a factor differing between independent groups of participants, which is not the case in the current design. We acknowledge that the term “repeated” suggests a time continuum, but time is not a requirement for a nested analyses. Based on your valuable comment we elaborated on this in l. 271-276.

Results -

301-307 – Please provide the results for the overall regression model. Was the overall model significant? This is critical for interpreting the rest of your findings. In other words, if the overall model was not significant, observing individual predictors that were significant (e.g., education) does not really matter.

 Our response: Thank you for this valid point and the possibility to explain this further. We have clarified this as the predictors were entered in the repeated measures ANCOVA (as between-subject factors) to explore whether the preferences for explanations differed based on these factors; we have now underlined the role of these variables a bit more in the method section (l. 273-276) and result section (l.359, 382, 406) and named these factors ‘placebo correlates’ to avoid confusion with regression analysis. To clarify: only one regression model was conducted with placebo knowledge as outcome and age, gender, education level, medication use, dispositional optimism, trait anxiety, general attitudes towards medication, and neuroticism as potential predictors. We have now also added more details on the significance of the model (l. 332-335).

The other models were repeated measures ANCOVAs in which the predictor variables age (now termed ‘placebo correlates’ to avoid confusion with regression/covariates): gender, education level, dispositional optimism, trait anxiety, neuroticism, placebo knowledge, and attitudes towards medication were entered as between-subject factors to explore the role of these variables in the outcomes of the three models (preference scores, perceived efficacy and willingness to participate).

328-381- The authors describe results of three different regression models presumably examining predictors of preference, perceived efficacy, and willingness to participate, but these models are not described in the statistical analysis section. What kind of regression model was used? How was the dependent variable score treated? For each model, was the score a sum total from the various items used to characterize preference, perceived efficacy, and willingness to participate? This will definitely need more detail.

 Our response: The models were described under ‘placebo explanations’ in the method section (l.265). All three outcomes (preference scores, perceived efficacy and willingness to participate) were used as separate sets of 8 numerical outcomes in a repeated measures ANCOVA, and the placebo correlates of within-subject differences between the 8 numerical outcomes were entered in the model as between-subject variables. Because perceived efficacy and willingness to participate were measured on a Likert-scale (therefore, ordinal), these outcomes were first converted into numerical values with a CATPCA (non-linear principal component analysis), in order for these outcomes to be entered in a two separate repeated measures ANCOVAs. The analyses were advised and based on the expertise of the co-author and statistician R.C.A. Rippe.

329-338- Please provide the results for the overall regression model in addition to your summary of individual predictors. Was the overall model significant?

 Our response: We kindly refer to our previous answer to this question (see above)..

351-362- Please provide the results for the overall regression model in addition to your summary of individual predictors. Was the overall model significant?

 Our response: We kindly refer to our previous answer to this question (see above)..

372-381- Please provide the results for the overall regression model in addition to your summary of individual predictors. Was the overall model significant?

 Our response: We kindly refer to our previous answer to this question (see above).

Discussion-

444-446- The investigator appears to interpret “Placebos can also cause side effects” as knowledge about nocebo effects. Indeed, information about nocebo effect knowledge is very important, but I am not entirely convinced that is what they measured with this item and would urge the authors to exercise some caution with referring to it as such. How do we know the respondents are assigning a negative connotation to the term “side effects”? It is unfortunate that this was not spelled out more clearly for study participants (e.g., Placebos can also cause negative or harmful effects).

 Our response: This is a very important and valid remark. We distributed this questionnaire in Dutch where side effects are called ‘bijwerkingen’, which translates to an unambiguous negative form of side effects. Adverse effects would have been a more appropriate translation, and therefore we have now adjusted this in the statements and the translation of the survey (l. 327) and in S5_File (English translation of the placebo questionnaire). 

444-447 – There’s hardly any discussion of the regression model that explored predictors of placebo quiz performance and found 13.6% variance explained. Was the amount of variance explained lower or higher than the authors expected? What other factors could account for the large amount of variance that was not explained by the model?

 Our response: Because this was one of the first studies to relate factors formerly associated with placebo effects to placebo knowledge, no specific hypotheses about the percentages of the explained variance were formed in advance (l.137-141). The results of our predictor analysis tell us that factors previously associated to contribute to placebo effects, do not explain the same amount of variance in placebo knowledge. We did find that education level was positively associated with placebo knowledge, perhaps medical information or experience with placebo effects would explain more variance in placebo knowledge, however further research is warranted. 

456-474 – I found the exploration and discussion of the underlying factor structure to be rather distracting and I am not sure this adds much value to the paper. In your revisions, please either make this more apparent to the reader or consider removing altogether.

 Our response: The rationale for integrating a principal component analysis was to explore whether participants perceived these explanations as the same underlying construct. We have now explained this in more detail (l. 288-294). We considered this analysis to be of importance in formulating placebo information strategies because a principal component analysis can serve as an exploration of the homogeneity of the reflection of internal beliefs about the underlying construct using the different formulations. To clarify this, we have also addressed this in the discussion section (l. 492-495).

479-480 – “Although the sample was rather large (377 completers)…”. Relative to which specific studies?

 Our response: We have now added references of studies with previous sample sizes, for example with healthy volunteers (Pugh et al; N=200), or patients (Chen & Johnson, N=211) (l. 161-164)

486-489 – Despite listing this as one of their aims, there’s also very little discussion of how exactly their results have extended prior work examining attitudes and acceptability towards placebos in treatment settings. I think this could be a separate paragraph rather than squeezed in as an afterthought in it’s present place in the manuscript.

 Our response: We thank you for this valuable point. We have now elaborated on this in the introduction (shortly: l. 141). In the method section, we now added a rationale for why we integrated different methods of placebo use with a short theoretical background (l. 220-226).

Limitations/Future directions section –

The discussion of the study limitations is pretty thin. Please add a paragraph or two on the potential shortcomings of this study to help the reader contextualize the findings. A few things that come to mind are:

(1) The cross-sectional nature of the study (e.g., what if attitudes change over time as new experiences occur (e.g., clinical trial participation, patient-provider interactions?). This is especially relevant considering how young your sample is.

 Our response: Thank you for your valuable input. We agree with your remarks and have addressed this limitation in the discussion (l. 532-535)

(2) How might various populations with chronic mental or physical health conditions benefit from this information? Is the present data sufficient for these groups or is more data directly collected from patient populations needed?

 Our response: Thank you for your valuable input. We have chosen this group because this represents a large part of society, so that the insights that this study brings forth may be broadly implemented. In future research it would be insightful to investigate whether the study findings can be replicated in patient samples or samples with health care professionals (l. 533-535).

(3) Is there anything to be learned from applying this research question/approach to nocebo effects and more thoroughly characterizing knowledge of nocebo effects than what was done here?

 Our response: Thank you for your valuable input. We agree that future information strategies for nocebo effects are of importance, especially since negative expectations can have an adverse effect on treatment outcomes. The present study may contribute to this by the information strategies proposed that also play a role in nocebo effects (e.g. conditioning) (l. 545-547)

(4) Just because a participant prefers a given strategy does not necessarily mean this translates to the largest placebo effects. This of course is an empirical question that could be tested in an experimental design and merits some thought/discussion about how such a study could be conducted.

 Our response: Thank you for your valuable input. We agree with your remarks and have addressed this in the manuscript, for example by investigating in future research whether optimized placebo strategies also result in larger placebo effects (l.543-546).

To be clear, these are just some potential ideas worth considering and I’m not saying that each of these needs to be added as limitations. Rather, I would like to see you give some thoughtful consideration to what you think are the primary limitations and discuss them accordingly.

Our response: Given this valuable input, we have made several changes in the manuscript regarding limitations to the sample, direction for future research, and discussed the implications of our results.

Reviewer #4: The authors conducted an online survey to examine knowledge and attitudes of the general population towards placebos, placebo effects, and placebo mechanisms. Results indicate that knowledge about placebo effects is generally high, with better knowledge in well-educated participants. Furthermore, the participants preferred explanations of placebo effects based on brain mechanisms and expectations, and preferences varied with age, education level, optimism, trait anxiety, and placebo knowledge. Perceived efficacy of the 8 proposed explanations showed comparable associations, while the willingness to participate varied with trait anxiety and attitudes towards medication. PCA revealed a two-dimensional structure for perceived efficacy and willingness to participate, separating between more positively stated mechanisms (such as expectations/mind-body/trust/brain mechanisms), and more negatively stated ones (such as transparency/neutral explanation). The use of placebos in clinical practice was widely accepted.

The study is well designed and reported, the statistical analysis is sound and the results support the conclusions. The findings are important as they provide various approaches for implementing placebo effects in clinical practice. The following issues should be addressed before publication:

Our response: We thank the reviewer for the compliments on our study. We addressed the questions and concerns below. 

- The sample comprises mainly young well-educated people. This is a major limitation of the results, since older and less-educated people may have different knowledge and attitudes about placebos and placebo effects. The authors discuss this issue very shortly. The implications are important and far-reaching and need to be addressed much more carefully by the authors. Furthermore, details of recruitment need to be explained in more detail (e.g., to whom the emails were sent, which social media platforms were used, etc.), and a rationale should be provided why the authors did not chose a more representative sample.

 Our response: Thank you for your valuable input. We have addressed your concerns by adding more nuances to the generalizability to clinical implementation in the title (by removing ‘for clinical implementation’), introduction and discussion (by rephrasing that this study provides insights for the development of placebo information strategies for clinical practice, instead of optimizing placebo information strategies for clinical practice). Moreover, we have further elaborated on the social media platforms that were used (l. 171) and the limitations of our sample (l. 520-541) 

- The title is very vague. It should include the type of study as well as the restriction to younger adults.

 Our response: Thank you for this insightful remark. We have now adjusted this by including ‘online survey’ in the title, and dropped ‘for clinical practice’. The restriction to younger adults in the title is in our opinion not entirely representative of the sample as we included people from varying ages (range: 16-78) although the median age was indeed younger as is also clearly described in Table 1 and the text.

- In Table 2, the entries for the first items should be double-checked; numbers appear to be reversed.

 Our response: Thank you for this remark, we appreciate your accuracy and have corrected this in Table 2 (l. 327).

---

## [Decision Letter · Decision Letter 1]

2 Feb 2021

Explaining placebo effects in an online survey study: Does ‘Pavlov’ ring a bell?

PONE-D-20-16802R1

Dear Dr. Smits,

We’re pleased to inform you that your manuscript has been judged scientifically suitable for publication and will be formally accepted for publication once it meets all outstanding technical requirements.

Kind regards,

Jenny Wilkinson, PhD

Academic Editor

PLOS ONE

Reviewers' comments:

Reviewer's Responses to Questions

**Comments to the Author**

1. If the authors have adequately addressed your comments raised in a previous round of review and you feel that this manuscript is now acceptable for publication, you may indicate that here to bypass the “Comments to the Author” section, enter your conflict of interest statement in the “Confidential to Editor” section, and submit your "Accept" recommendation.

Reviewer #1: All comments have been addressed

Reviewer #3: All comments have been addressed

Reviewer #4: All comments have been addressed

2. Is the manuscript technically sound, and do the data support the conclusions?

Reviewer #1: (No Response)

Reviewer #3: Yes

Reviewer #4: Yes

3. Has the statistical analysis been performed appropriately and rigorously? 

Reviewer #1: (No Response)

Reviewer #3: Yes

Reviewer #4: Yes

4. Have the authors made all data underlying the findings in their manuscript fully available?

Reviewer #1: (No Response)

Reviewer #3: Yes

Reviewer #4: Yes

5. Is the manuscript presented in an intelligible fashion and written in standard English?

Reviewer #1: (No Response)

Reviewer #3: Yes

Reviewer #4: Yes

6. Review Comments to the Author

Reviewer #1: (No Response)

Reviewer #3: Thank you for your thoughtful responses to my concerns. I appreciate the attention to detail and have no further comments.

Reviewer #4: (No Response)

7. PLOS authors have the option to publish the peer review history of their article (what does this mean?). If published, this will include your full peer review and any attached files.

Reviewer #1: **Yes: **Pekka Louhiala

Reviewer #3: No

Reviewer #4: No

---

## [Editor Report · Acceptance letter]

17 Feb 2021

PONE-D-20-16802R1 

Explaining placebo effects in an online survey study: Does ‘Pavlov’ ring a bell? 

Dear Dr. Smits:

I'm pleased to inform you that your manuscript has been deemed suitable for publication in PLOS ONE. Congratulations! Your manuscript is now with our production department. 

Kind regards, 

on behalf of

Dr Jenny Wilkinson 

Academic Editor

PLOS ONE